# Persistence and Transfer of Foodborne Pathogens to Sunflower and Pea Shoot Microgreens during Production in Soil-Free Cultivation Matrix

Wenjun Deng [1], Gina M. Misra [2], Christopher A. Baker [1] and Kristen E. Gibson [1,*]

[1] Department of Food Science, Division of Agriculture, University of Arkansas, Fayetteville, AR 72704, USA; wd001@uark.edu (W.D.); cab010@uark.edu (C.A.B.)
[2] Blue Marble Space Institute of Science, 600 1st Avenue, 1st Floor, Seattle, WA 98104, USA; ginamariemisra@gmail.com
* Correspondence: keg005@uark.edu; Tel.: +1-479-575-6844

**Abstract:** Microgreens are an emerging salad crop with properties similar to those of sprouted seeds and lettuce. This study aimed to determine bacterial pathogen persistence during microgreen cultivation and transfer from soil-free cultivation matrix (SFCM) to mature microgreens. *Salmonella enterica* subsp. *enterica* ser. Javiana and *Listeria monocytogenes* were inoculated onto biostrate mats as well as peat SFCM and sampled (day 0). Next, sunflower and pea shoot seeds were planted (day 0) and grown in a controlled environment until the microgreen harvest (day 10). On day 10, SFCM and microgreens were sampled to determine the pathogen levels in the SFCM and the pathogen transfer to microgreens during production. *Salmonella* Javiana log CFU/g were significantly higher than *L. monocytogenes* in SFCM on day 10 in both planted and unplanted regions ($p < 0.05$). Significant differences in pathogen transfer (log CFU/g) were observed between the pea shoot and sunflower microgreens, regardless of the pathogen or SFCM type ($p < 0.05$). Meanwhile, pathogen transfer to the pea shoot and sunflower microgreens from the biostrate was 1.53 (95% CI: −0.75–3.81) and 5.29 (95% CI: 3.01–7.57) mean log CFU/g, respectively, and transfer from the peat was 0.00 (95% CI: −2.28–2.28) and 2.64 (95% CI: 0.36–4.92) mean log CFU/g, respectively. Results demonstrate that pathogen transfer to microgreens during production is influenced by SFCM and microgreen variety.

**Keywords:** microgreens; peat; vermiculite; growing mats; *Salmonella* spp.; *Listeria monocytogenes*; sunflower; pea shoot

## 1. Introduction

Microgreens are a raw salad product similar to sprouted seeds and lettuce. However, while there are similarities, some aspects of microgreen production differ from that of sprouts and lettuce. While sprouts are germinated for five days in a warm, mostly enclosed, moist environment [1,2], microgreens are germinated for up to 72 h in either a hydroponic nutrient solution, soil, or a soil substitute [3–5]. Germinated microgreens are then allowed to grow for approximately 10 to 20 days—until the opening of the cotyledon or the formation of the first set of true leaves. Lettuce, by contrast, is typically grown in a field, or hydroponically, and reaches the mature rosette stage after 90 days [6]. When produced as a "baby" variety, lettuce may also be cultivated in greenhouses and harvested at 38 to 43 days [7].

Microgreens are a profitable and popular crop for production in controlled environment agriculture (CEA) systems [8–12]. As such, microgreen growing operations comprise a notable portion of the CEA industry. Importantly, the production environment and conditions under which leafy greens are grown may influence the plant's uptake of bacteria, which include human pathogens that contribute to produce-associated foodborne illness [13,14]. Thus far, microgreens have not been implicated in any known illnesses

or outbreaks but have been the subject of at least 7 product recalls since 2016 in the US and Canada due to possible contamination with *Salmonella* spp. and *Listeria monocytogenes* [15–21]. Human and plant pathogens are known to utilize the plant root system to gain access to internal plant tissues, which makes post-harvest washing with sanitizers ineffective [14,22]. Therefore, studying the aspects of leafy green production that may increase the risk of contamination via the cultivation matrix is a necessary preventive strategy.

Pathogen uptake into leafy greens from soil has been extensively studied, although these studies are primarily limited to laboratory-controlled investigations [22–34]. In addition to soil, several studies have explored pathogen uptake by hydroponically grown crops [35–40]. Pathogen internalization may be affected by several factors including the pathogen type, the growth substrate used, the moisture level near the rhizosphere, the contamination route (seed, irrigation), and the plant variety [26,30,37,41,42]. Additional internalization routes, such as through stomata as well as wounds and cuts, have also been characterized [27]. The July 2021 recall of hydroponically grown lettuce and the subsequent multistate outbreak in the U.S. due to contamination with the *Salmonella* serotype Typhimurium illustrate the potential food safety risks associated with CEA-grown leafy greens [43].

Previous work investigating foodborne pathogens within microgreen growing systems have examined peat-based germination mixes [44–46], peat/peat–perlite [2,47], natural fiber mats [48], synthetic fiber mats [2,46,48], and hydroponic nutrient solutions [49]. Variations in nutrient quality and chemical composition exist between soil-free cultivation matrices used (SFCM) [2,50]. Moreover, the SFCM type has been shown to impact pathogen survival over time [50]. However, the impact of SFCM on pathogen survival during microgreen production has not been extensively studied.

Microgreen varieties studied in a food safety context have included radish [45–47], cabbage [51], broccoli [48], kale [49,52], mustard [49], rapini [2], lettuce [47], swiss chard [44], and herb varieties [48]. However, based on a survey of microgreen growing operations selling within the United States, previously studied plant varieties have not included two of the most commonly grown microgreens—sunflower and pea shoots [10]. In addition, previous studies on microgreens and pathogen contamination risks have primarily focused on Shiga toxin-producing *Escherichia coli* and, to a lesser extent, *L. monocytogenes* and select *S. enterica* serotypes [11]. A variety of *Salmonella enterica* serotypes have been attributed to fresh produce contamination including *S.* Javiana—the fourth most common, non-typhoidal *Salmonella* serotype in the U.S. [53]. Therefore, the present study aimed to determine *S.* Javiana and *L. monocytogenes* persistence in a microgreen cultivation system and in the transfer from soil-free cultivation matrix (SFCM) to mature microgreens.

## 2. Materials and Methods

### 2.1. Bacterial Culture Preparation

*Listeria monocytogenes* strain F2365 (FSL R2-574) and *Salmonella enterica* subsp. *enterica* serotype Javiana (ATCC BAA1593; ATCC, Manassas, VA, USA) were inoculated in 10 mL of a brain heart infusion (BHI; BD, Franklin Lakes, NJ, USA) and tryptic soy broth (TSB; Difco, Sparks, MD, USA) and incubated for 24 h with shaking (100 rpm) at 37 °C. The cultures were centrifuged at $4000\times g$ at 4 °C for 10 min, the supernatant was discarded, and the cells were washed with 10 mL of sterile $1\times$ phosphate buffered saline (PBS, pH = 7.4). Centrifugation and cell washing were repeated. Following a third centrifugation step, the cells were resuspended in 10 mL of PBS, vortexed for 10 s, and 1 mL of each culture was combined to obtain a bacterial cocktail. The cells were diluted to a final concentration of 7 log CFU/mL in PBS. To confirm the bacterial concentrations added to the SFCM, the bacterial inoculum containing *L. monocytogenes* and *S.* Javiana was diluted and plated on a selective agar prepared with Oxford Medium (Difco) which was supplemented with Oxford Listeria Selective Supplement (Millipore Sigma, St. Louis, MO, USA) (MOX), and a xylose lysine tergitol-4 base (Hardy Diagnostics, Santa Maria, CA, USA) which was

supplemented with XLT4 supplement (Hardy Diagnostics) (XLT4), respectively. Plates were incubated at 35 °C for 48 h for MOX and 37 °C for 24 h for XLT4.

## 2.2. Preparation of Experimental Set-Up and Day 0 Sampling

A single 3.5 mm thick BioStrate® Felt biopolymer, a natural fiber blend mat (Quick Plug North America, South Portland, ME) (biostrate) and a Jiffy-Mix Soil-less Starter Peat/Vermiculite mix (Harris Seeds, Rochester, NY) (peat) were added to separate no-drainage trays (54 × 28 × 6.4 cm) (Harris Seeds) and watered with sterile deionized (DI) water. The peat was mixed with gloved hands to distribute the water homogenously. Once the bacterial cocktail inoculum was prepared, as described in Section 2.1, 50 mL PBS with 7 log CFU/mL of the bacterial inoculum was evenly distributed over each SFCM. An equal volume of PBS was added to the negative control SFCM trays. Day 0 SFCM samples were immediately obtained following inoculation (two samples per tray area, six total samples per tray). The biostrate SFCM was sampled by cutting 2.5 × 2.5 cm squares (equivalent to approximately 5 g) out of the mat using sterile forceps and scissors. Approximately 5 g of peat was sampled with a sterile metal scoop. Bacteria were recovered by placing the SFCM samples in 50 mL sterile centrifuge tubes containing 10 mL PBS, vortexing them for 1 min at maximum speed in 15 s pulses, diluting them with PBS, and plating them on MOX and XLT4 agar followed by their incubation as described in Section 2.1. Following dilution and plating, the SFCM samples were then dried in an oven at 80 °C for 24 h to determine the sample dry weight. The SFCM dry weights were used to report log CFU/g. Based on the range of SFCM weights following drying, the limit of detection (LOD) from the SFCM for both pathogens ranged from 2.2 to 2.6 log CFU/g. To determine the background total aerobic plate counts (APC) from the SFCM prior to their use in experiments, biostrate and peat (1.0 g) were separately eluted in 10 mL of PBS, vortexed for 10 s, and incubated for 1 h at 21 °C. Following the 1 h incubation, samples were vortexed for 10 s, diluted in PBS, plated on tryptic soy agar (TSA) (Difco), and incubated at 25 °C for 48 h. The APC LOD was 1 log CFU/g.

## 2.3. Germination and Cultivation of Microgreens on SFCM

Prior to their planting, 30 g of black oil sunflower (*Helianthus annuus*) seeds (Tiensvold Farms, Rushville, NE, USA) and 50 g of pea shoot (*Pisum sativum*) seeds (Johnny's Selected Seeds, Winslow, ME, USA) were soaked for 6 h in separate sterile beakers containing 500 mL of sterile DI water. Different weights were used for each seed variety, due to the seed size and respective surface area on the SFCM. The seed soaking procedure was performed separately for each tray. Once the SFCM was prepared and sampled on day 0 (see Section 2.2), the soaked seeds were drained in a sterile wire mesh strainer and spread evenly over their respective SFCM areas (Figure 1). Undiluted, seed-soaked water was plated on MOX and XLT4 selective agar to determine pathogen presence and incubated as described in Section 2.1. A separate analysis was performed to determine the amount of water retained by each seed variety during the 6 h soak. Seeds were weighed and then soaked. After soaking, the seeds were collected in a wire mesh strainer to remove excess water and then weighed again. The water retention was calculated by dividing the difference between the pre- and post-soak seed weight by the final seed weight and is represented as the percentage of final seed weight that was due to water. Aerobic plate counts from each seed type were determined by diluting seed soak water in PBS, plating on TSA, and incubating at 25 °C for 48 h with a LOD of 1 log CFU/g.

For planting, the seeds were separated by an unplanted area in the middle of the tray (Figure 1). Following planting, the seeds were covered for two days with a second germination tray that had been decontaminated with 70% ethanol and inverted to form a lid to retain moisture and promote seed germination in near darkness. The seeds were misted with sterile DI water four to five times throughout the day. On day 2 of germination, the tray covers were removed, and the microgreens were grown to maturity under a 16 h photoperiod using two GrowBright 4-foot T5 6400K (5000 lumens) Compact Fluorescent

Lamps (HTG Supply, Callery, PA, USA) for eight subsequent days (10 days total). The SFCM were monitored periodically throughout each day and irrigated with sterile DI water. Based on the preliminary cultivation trials and visual inspection, approximately 200 and 300 mL of sterile DI water was needed per day for the biostrate and peat, respectively. During irrigation, sterile DI water was poured directly over the SFCM, and the trays were gently rocked to distribute the water evenly. A 5.5 L humidifier (HAUEA, Shenzhen, China) was used to maintain the growing environment at a target relative humidity (RH) of 40–60%. The RH and temperature were monitored with an indoor temperature and humidity monitor (AcuRite, Lake Geneva, WI, USA) and a HOBO® Bluetooth Low Energy Temperature/Relative Humidity Data Logger (Onset Computer Corporation, Bourne, MA, USA).

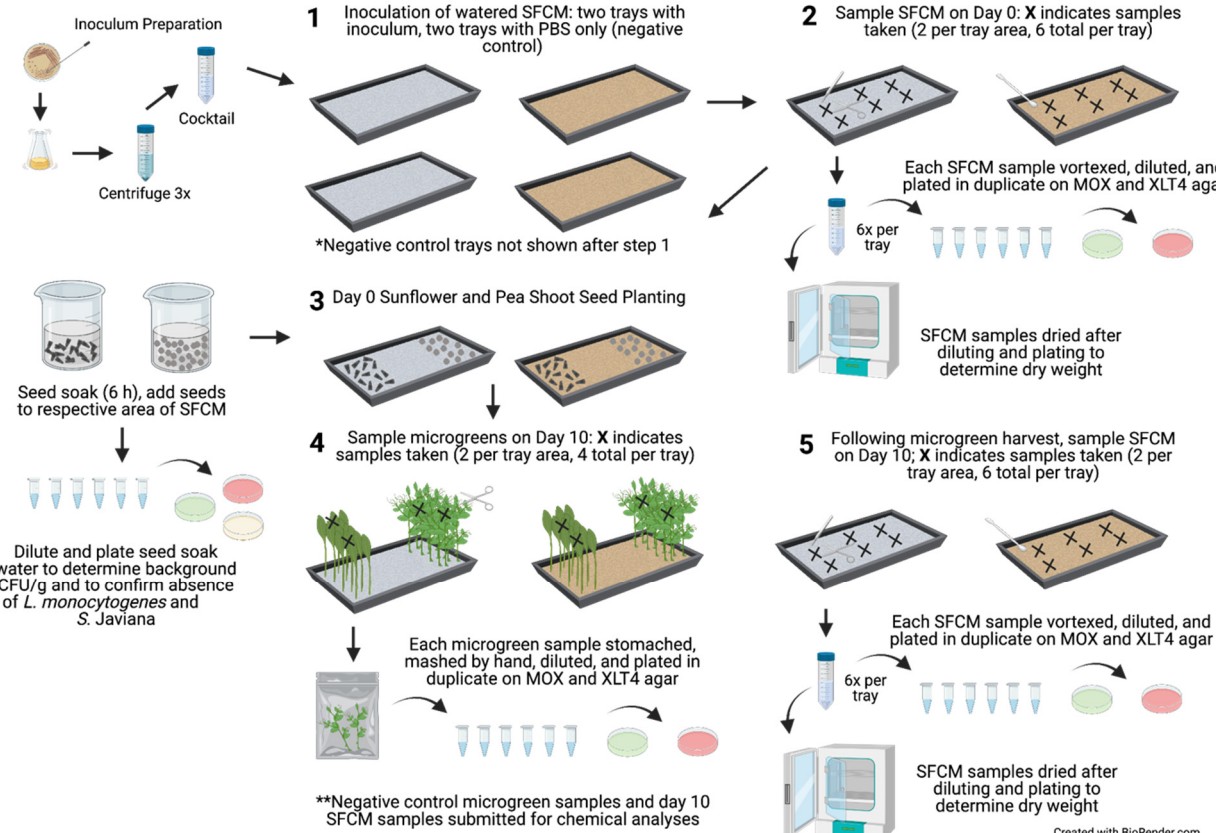

**Figure 1.** Flow diagram of experiment. Inoculum preparation, SFCM inoculation, day 0 SFCM sampling, seed soaking and planting, and day 10 SFCM and microgreen sampling. The same tray set up was performed simultaneously for the negative control trials, in which the SFCM were inoculated with 50 mL of PBS in each tray. Gray and brown trays indicate biostrate and peat SFCM, respectively. Created with Biorender.com, accessed on 25 October 2021.

### 2.4. Day 10 SFCM and Microgreen Sampling

Following 10 days of microgreen cultivation, the SFCM were sampled as described in Section 2.2. Following the SFCM sampling, four to eight microgreens (2 to 3 g per sample) were sampled by gently holding the top of the microgreens with sterile gloved hands, cutting to 2 cm above the SFCM with sterile scissors, and placing samples in Whirl-Pak bags (two samples per microgreen type, four total samples per tray). Microgreen sample fresh weights were determined, and 10 mL of PBS was added to each bag and stomached for 2 min at 200 rpm (Stomacher 400 Circulator; Seward, Worthing, UK), followed by crushing the plant tissue by hand for 30 s. The microgreen and SFCM samples were diluted in PBS, plated on MOX and XLT4 agar, and incubated as described in Section 2.1. Section 2.2 to Section 2.4, outlining experimental set-up and sample collection, were repeated twice for a total of two experimental trials, with all samples analyzed in technical duplicates. The

LOD for microgreen analysis in trials 1 and 2 was 1.76 log CFU/g and 0.74 log CFU/g of fresh weight, respectively. The LOD differed due to the adjustments in plating volume between trials. Levels below the LOD were assigned values of 0 log CFU/g for both trials. Notably, there were no instances in which bacterial concentrations in the microgreens were above 0.74 log CFU/g but below 1.76 log CFU/g in trial 2; thus 0 log CFU/g values for each trial were considered similar despite the differences in LOD between trials.

### 2.5. Chemical Analysis of SFCM and Microgreens

The soil-free cultivation matrix and the microgreen samples from the negative control trays were analyzed at the University of Arkansas' Agricultural Diagnostic Laboratory for total carbon, total nitrogen, nitrate–nitrogen ($NO_3$-N), and minerals. Total carbon and nitrogen were determined by combustion [54], $NO_3$-N was determined by UV-Vis spectroscopy following deionized water extraction via the modified Cataldo method [55], and the minerals were analyzed via the Melich-3 method by inductively coupled plasma mass spectrometry following a saturation extraction [56]. The pH and electrical conductivity of SFCM were determined at a 1:2 SFCM to water ratio as described by Sikora and Kissel [57], and Wang et al. [58], respectively. Ammonium–nitrogen ($NH_4$-N) was determined by a 2 mol $L^{-1}$ KCl extraction [59]. The peat samples were submitted for analysis in their original state, and biostrate samples were cut into approximately $2.5 \times 2.5$ cm squares prior to their submission. If present, microgreen roots were not removed from the SFCM samples prior to analysis, and no correction for moisture values was made prior to the SFCM sample analysis. A homogenous distribution of moisture and plant roots across the mats was assumed due to the size of the square cut from the mats. The microgreen plant samples were dried at 55 °C and ground through a 1 mm sieve prior to analysis. The microgreen samples were then evaluated for total C, total N, $NO_3$-N, and minerals as described for SFCM. An acid detergent fiber (ADF) analysis measuring cellulose and lignin, and a neutral detergent fiber (NDF) analysis measuring hemicellulose, cellulose, and lignin, were performed on microgreen plant samples by Ankom digestion [60,61].

### 2.6. Statistical Analysis

Bacterial concentrations were $log_{10}$ transformed prior to the statistical analyses. A split-split-split plot with a randomized complete block design was analyzed in the mixed-effect model to compare the mean log CFU/g values across the SFCMs (block) used, based on the plant type, day, and pathogen type. A split-split plot with a randomized complete block design was analyzed in the mixed-effect model to compare the mean log CFU/g values across the plant types (block) used, based on SFCM type and pathogen type. Both of the mixed-effect models were robust and accounted for the unequal variation of groups. The least-squares means were calculated to compare the mean log CFU/g values based on the factors included in each analysis. Mean values were compared with the Tukey's honest significant difference test with a significance level of 0.05. All statistical analyses were performed using R version 3.6.2 (http://www.R-project.org, accessed on 1 September 2021).

## 3. Results

### 3.1. Relative Humidity and Temperature during Microgreen Cultivation

The average microgreen cultivation environment temperature in trials 1 and 2 during the 10 day cultivation period was 20.8 and 20.4 °C, respectively. Temperatures ranged from 18.9 to 22.2 °C in trial 1, and 17.8 to 22.5 °C in trial 2. The average relative humidity of the growing environment in trials 1 and 2 during the 10 day cultivation period was 45.2 and 52.1%, respectively, with a range of 21 to 69% in trial 1 and 24.2 to 72.5% in trial 2. The percentage of the final seed weight post 6 h soak was 43 and 42% water for the sunflower and pea shoots, respectively.

### 3.2. Pathogen Persistence in Planted and Unplanted SFCM

Following the inoculation and immediate sampling of SFCM on day 0, consistent concentrations of *S.* Javiana and *L. monocytogenes* were observed in biostrate with a mean log CFU/g of 6.24 ± 0.10 and 6.33 ± 0.07, respectively, while larger variations among samples were observed in peat at 5.33 ± 1.10 and 5.34 ± 1.10, respectively. There was no significant difference between *S.* Javiana and *L. monocytogenes* concentrations in SFCM on day 0 ($p > 0.05$).

No significant differences in pathogen concentrations were observed based on the SFCM type or the microgreen root presence (sunflower, pea shoot, or unplanted) on day 10 ($p > 0.05$). When pathogen levels were averaged across both SFCM types in the presence of the sunflower and pea shoot microgreens, as well as in unplanted SFCM, there was a significant difference in the mean log CFU/g between pathogen types on day 10, with higher levels of *S.* Javiana (6.99, 95% CI: 3.76–10.22) in comparison to *L. monocytogenes* (4.78, 95% CI: 1.55–8.01) ($p < 0.05$). By day 10 of the microgreen cultivation, an increase in the *S.* Javiana mean log CFU/g was observed in biostrate SFCM, and minimal differences in *S.* Javiana were observed in peat, regardless of the microgreen presence (sunflower, pea shoot, or unplanted). This trend was not observed for *L. monocytogenes* in peat (Figure 2). *Listeria monocytogenes* concentrations were maintained during the 10 day cultivation period in biostrate regardless of the microgreen presence. In peat, minimal changes in *L. monocytogenes* populations were observed over the 10 day period with sunflower microgreens, but populations had declined over 2 log CFU/g in the presence of pea shoots, as well as in unplanted peat. The lowest pathogen concentrations observed in SFCM on day 10 were in unplanted peat, where 5.33 ± 0.48 and 2.45 ± 0.03 log CFU/g *S.* Javiana and *L. monocytogenes* were recovered, respectively (Figure 2).

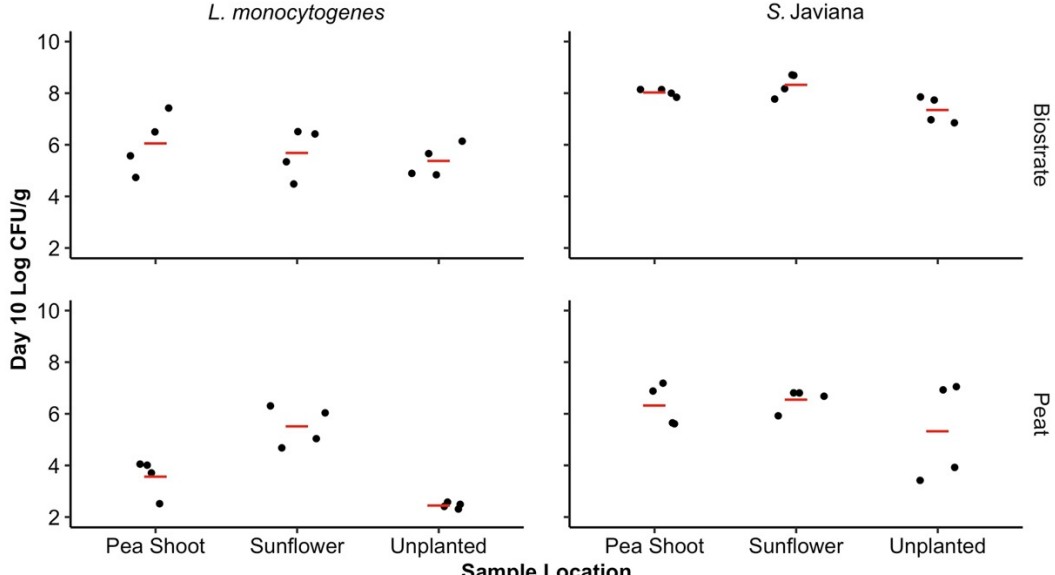

**Figure 2.** Concentration of bacterial pathogens in SFCM on day 10. The observed log CFU/g of *Salmonella* Javiana and *L. monocytogenes* on day 10 in biostrate and peat SFCM, influenced by sunflower, pea shoot microgreen root presence, or absence [unplanted]. Significant differences were observed between the *S.* Javiana and *L. monocytogenes* log CFU/g regardless of SFCM or microgreen presence ($p < 0.05$). Based on the range of the SFCM weights following drying, the LOD from SFCM for both pathogens ranged from 2.2 to 2.6 log CFU/g. Concentrations of *S.* Javiana and *L. monocytogenes* in SFCM on day 0 were of a mean log CFU/g of 6.24 ± 0.10 and 6.33 ± 0.07 in biostrate and 5.33 ± 1.10 and 5.34 ± 1.10 in peat, respectively. Mean log CFU/g for each growth media/pathogen combination is represented by the red line.

Lower background aerobic bacteria concentrations (mean log CFU/g) were observed in biostrate (0.24 ± 0.34) when compared to peat (4.43 ± 0.43). The pea shoot and sunflower-soaked seeds had background aerobic plate count concentrations of 0.95 ± 0.92

and 5.27 ± 0.63 mean log CFU/g, respectively. No *Salmonella* spp. or *L. monocytogenes* colonies were recovered from the negative control SFCM types with or without a root presence (PBS-inoculated) on day 0 or day 10, for either trial.

### 3.3. Pathogen Concentrations in Microgreens Grown in SFCM

The pathogen levels were assessed for the microgreens to determine if the microgreen variety and SFCM had influenced pathogen transfer and persistence after 10 days of microgreen cultivation. The microgreen variety significantly influenced the pathogen levels recovered from harvested microgreens on day 10 for both the biostrate ($p$ = 0.005) and peat ($p$ = 0.009) SFCM. In biostrate, the mean log CFU/g among both pathogen types was 1.53 (95% CI: −0.75–3.81) for the pea shoots and 5.29 (95% CI: 3.01–7.57) for the sunflower microgreens, while the mean log CFU/g for the pea shoot and sunflower microgreens in peat was 0.00 (95% CI: −2.28–2.28) and 2.64 (95% CI: 0.36–4.92), respectively.

No significant differences were observed among the pea shoot ($p$ = 0.20) and sunflower ($p$ = 0.27) microgreens based on pathogen type. However, higher *S.* Javiana levels were recovered from the pea shoot and sunflower microgreens (1.27, 95% CI: −3.20–5.75 and 4.39, 95% CI: −0.08–8.86) in comparison to *L. monocytogenes* (0.26, 95% CI: −4.21–4.73 and 3.54, 95% CI: −0.93–8.01) (Figure 3). Similarly, the SFCM did not significantly influence pathogen levels in the pea shoot ($p$ = 0.10) or sunflower ($p$ = 0.06) microgreens. There were no *Salmonella* spp. or *L. monocytogenes* colonies recovered from either microgreen variety samples which were grown in the negative control biostrate and peat SFCM during harvest (day 10) in either trial.

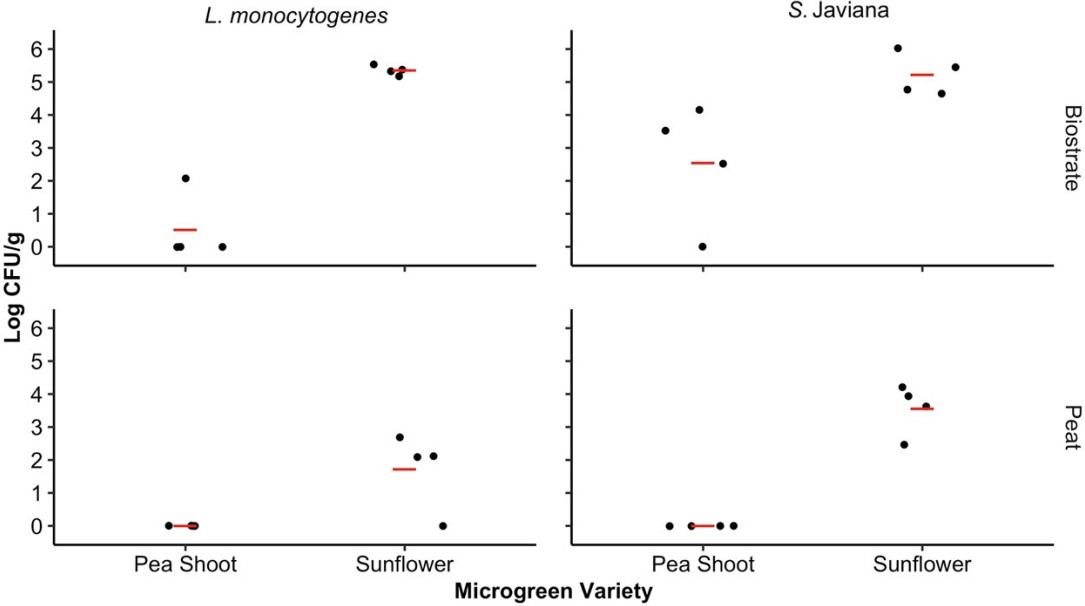

**Figure 3.** Concentrations of the bacterial pathogens in microgreens on day 10. The *Salmonella* Javiana and *Listeria monocytogenes* log CFU/g recovered from the sunflower and pea shoot microgreens grown in biostrate and peat on day 10. Significant differences were observed between the pea shoot and sunflower microgreens in both biostrate and peat SFCM ($p$ < 0.05). The LOD for microgreen analysis in trials 1 and 2 was 1.76 log CFU/g and 0.74 log CFU/g of fresh weight, respectively, and differed due to the adjustments in plating volume between trials. Concentrations below the LOD were assigned values of 0 log CFU/g for both trials. Mean log CFU/g for each treatment level is represented by the red line.

### 3.4. Chemical Analysis of Planted and Unplanted SFCM

The chemical composition of peat and biostrate SFCM in the presence and absence of microgreen roots was analyzed on day 10. The sunflower, pea shoot, and unplanted SFCM had a pH of 7.4, 7.6, and 7.0 for biostrate and a pH of 5.9, 6.0, and 5.7 for peat, respectively, on day 10 (Supplemental Table S1). Among the three types of peat SFCM

(sunflower, pea shoot, or unplanted), the electrical conductivity in the unplanted peat was 792 μmhos cm$^{-1}$, in comparison to the peat-grown sunflower and pea shoot at 250 and 272 μmhos cm$^{-1}$, respectively. Similarly, $NO_3$-N, K, Mg, and Na levels were considerably higher in the unplanted peat when compared to planted peat (Supplemental Table S1). The SFCM was analyzed on an as-is basis without the drying of the samples prior to analysis on day 10 for trial 2 only, thus the comparisons between biostrate and peat SFCM were not evaluated statistically.

*3.5. Chemical Analysis of Harvested Sunflower and Pea Shoot Microgreens*

On day 10, the microgreens were harvested and submitted for chemical analysis, which revealed several differences between the sunflower and pea shoot microgreens cultivated in both biostrate and peat. When comparing the fiber values between microgreen varieties, the sunflower microgreens had a 2.1- and 1.64-fold higher percentage of ADF and a 1.99- and 1.70-fold higher percentage of NDF when compared to pea shoot microgreens in biostrate and peat, respectively. A higher percentage of nitrogen levels was observed in the pea shoot microgreens when grown in both SFCMs (Table 1). The sunflower microgreens grown in biostrate exhibited a sodium (Na) level of 3312 mg/kg, which was a considerably higher level than in the biostrate-grown pea shoot microgreens (663 mg/kg), peat-grown sunflower microgreens (388 mg/kg), and peat-grown pea shoot microgreens (342 mg/kg). Lastly, $NO_3$-N was highest in the peat-grown microgreens with 4550 mg/kg and 1925 mg/kg for the sunflower and pea shoots, respectively, in comparison to 375 mg/kg and 125 mg/kg for the biostrate-grown microgreens (Table 1). Statistical analysis was not performed on the chemical analysis data as results are presented from trial 2 microgreens only.

**Table 1.** Chemical composition analysis of sunflower and pea shoot microgreens harvested on day 10 (trial 2 only).

| | Biostrate | | Peat | |
|---|---|---|---|---|
| **Analyte [1], Unit** | **Sunflower** | **Pea Shoot** | **Sunflower** | **Pea Shoot** |
| % ADF | 26.16 | 12.43 | 24.79 | 15.12 |
| % NDF | 28.95 | 14.52 | 27.96 | 16.43 |
| % Nitrogen | 3.56 | 7.65 | 4.21 | 8.31 |
| % Carbon | 50.49 | 43.71 | 46.23 | 42.78 |
| % P | 0.74 | 0.73 | 0.73 | 0.81 |
| % K | 1.18 | 1.83 | 2.15 | 2.43 |
| % Ca | 0.101 | <0.001 | 0.430 | 0.162 |
| % Mg | 0.42 | 0.22 | 0.80 | 0.34 |
| % S | 0.20 | 0.24 | 0.34 | 0.37 |
| Na (mg/kg) | 3312 | 663 | 388 | 342 |
| Fe (mg/kg) | 46 | 70 | 172 | 122 |
| Mn (mg/kg) | 20 | 11 | 120 | 60 |
| Zn (mg/kg) | 61 | 52 | 72 | 50 |
| Cu (mg/kg) | 21 | 15 | 25 | 13 |
| B (mg/kg) | 14 | 10 | 21 | 12 |
| $NO_3$-N (mg/kg) | 375 | 125 | 4550 | 1925 |

[1] Chemical analysis was determined on a dry (55 °C) plant concentration basis. Plant samples were ground and passed through a 1 mm sieve. ADF, acid detergent fiber test. NDF, neutral detergent fiber test. Units for each analyte are on the left-hand column; percent refers to the percentage dry weight of the sample.

## 4. Discussion

In the present study, the data confirm that *S.* Javiana and *L. monocytogenes* respond differently during microgreen cultivation in SFCM. Overall, pathogen survival was generally greater in biostrate compared to peat—both unplanted and planted—which is partially supported by previous work on SFCM alone [50]. Misra and Gibson [50] observed similar trends in pathogen persistence in peat and biostrate, with notable log reductions for *L. monocytogenes* in peat (>4 log CFU/mL) over a 10 day period when compared to biostrate

(<1 log CFU/mL). Meanwhile, *S.* Javiana maintained consistent concentrations in biostrate, and minimal log reductions were observed in peat (1 to 2 log CFU/mL) after 10 days.

Additional studies also support the observation that pathogen survival was influenced by the cultivation matrices [2,46,48]. Wright and Holden [48] characterized the ability of *E. coli* O157:H7 to colonize microgreens based on the cultivation matrix type and reported an increase of 2 to 4 log CFU/mL *E. coli* O157:H7 in SFCM after 7 days at 21 °C under static conditions, although concentrations also increased by 2 log CFU/mL in the control which contained minimal plant growth media. Similar to the current study, Misra and Gibson [50] prepared the inoculum in PBS, which is not supportive of bacterial growth, and thus, any observed increase in bacteria was due only to the cultivation matrix. Moreover, Wright and Holden [48] sterilized the growth matrices prior to use, which does not reflect real-world microgreen cultivation practices.

Xiao et al. [46] compared the survival of *E. coli* O157:H7 during radish microgreen cultivation in a peat-based germination mix and a synthetic fiber growing mat. The authors observed a significant difference in *E. coli* O157:H7 levels between the cultivation matrix type used following a 7 day growth period. Specifically, at high-level inoculation (5.6 log CFU/g), 3 to 4 log CFU/g remained in the peat-based mix, whereas *E. coli* O157:H7 populations were maintained in the growing mat after 7 days. The observed differences in pathogen levels in peat-based mixes and fiber-based growing mats across multiple studies may be due to the competition for nutrients between resident microorganisms found in peat that is not present in growing mats [2,50,62,63]. Baseline aerobic plate count of bacteria were nearly 20 times greater in the peat-based germination mix when compared to biostrate in the present study. Di Goia et al. [2] also demonstrated significantly higher levels of aerobic bacteria, yeast and mold, and *Enterobacteriaceae* in a peat-based mix when compared with three different growing mats. Previous works have shown that Canadian *sphagnum* peat moss can suppress numerous root rot pathogens, through competition for nutrients with resident microorganisms [64,65].

The chemical composition of the SFCM used may also be important to pathogen persistence; however, most studies do not characterize SFCM as shown in the present study. While there are some notable differences in the chemical compositions of SFCMs, e.g., greater concentrations of macrominerals (Mg, Ca, S), microminerals (Fe, Mn), and nitrate–nitrogen in peat when compared to biostrate (Supplemental Table S1), the variation across unplanted and planted regions and within microgreen varieties confounds our ability to discuss, with confidence, the potential influence of these analytes on pathogen survival and their subsequent transfer to the edible portion of the microgreen. However, previous work in conventional soil production of leafy greens suggests that activity in the plant rhizosphere influences the soil chemical profile [26,66]. Therefore, further investigations are needed to determine the critical factors of SFCM in the presence of microgreens which sustain pathogens during microgreen production.

The impact of the microgreen variety on pathogens in SFCM alone, and with root presence, was investigated in this study. While the transfer of pathogens to the edible product is one of the greatest food safety concerns, any differences in survival on SFCM with or without plant roots would suggest that the rhizosphere composition plays a role in the survival of foodborne pathogens in indoor microgreen cultivation systems [67]. In the present study, day 10 sampling revealed no significant differences in pathogen concentrations in SFCM, based on either microgreen variety or root presence (sunflower, pea shoot, or unplanted); however, it is also notable that the pathogen concentrations were lowest in the unplanted biostrate and peat on day 10 when compared to planted SFCM. Root exudates that are specific to each plant variety, as well as the organisms belonging to the root microbiome, may enhance or suppress the growth of *Salmonella* spp., *L. monocytogenes*, and other major foodborne pathogens, and should be further characterized [68].

Chitarra et al. [69] considered the impact of plant type on the persistence of *E. coli* O157:H7 and *L. monocytogenes* during the cultivation of herbs and baby salad plants. The authors inoculated a soil-less mixture (1:3 *v/v* peat–perlite) cultivation system via

contaminated irrigation water at a 5 log CFU/g final concentration and observed similar levels of *E. coli* O157:H7 after 20 days, with a large variation between *L. monocytogenes* levels. depending on the variety of the growth substrate. Interestingly, the growth substrate collected proximal to basil completely inactivated *L. monocytogenes* while significantly reducing *E. coli* O157:H7 when compared to other varieties [69]. Additionally, Cooley et al. [70] reported the inhibition of *S.* Newport and *E. coli* O157:H7 in the presence of *Enterobacter asburiae* on the roots of *Arabidopsis thaliana*. It was later reported that *E. coli* O157:H7 and epiphytic bacteria compete for the same carbon sources that are present in lettuce seedling exudates [71].

The microgreen variety was also compared based on the pathogen transfer to the edible portion. Regardless of the SFCM evaluated in this study, pathogen concentrations were significantly higher in harvested sunflower microgreens in comparison to pea shoot microgreens, which highlights plant variety differences that exist within the microgreen industry. Although the internalization of pathogens was not specifically determined here, previous studies have reported significant differences in bacterial pathogen internalization, depending on plant variety and cultivar [28,47,69,72–74]. Klerks et al. [73] identified specific interactions between *Salmonella* and root exudates of select lettuce cultivars which aided pathogen internalization. Meanwhile, Erickson et al. [28] suggested the role of various plant defenses (e.g., total phenols and antioxidant capacity chemicals) in preventing endophytic colonization by certain human pathogenic bacteria via initial foliar contamination. Specific to microgreens, Işik et al. [47] compared the transfer of *E. coli* O157:H7 to the edible portion of two microgreen varieties (lettuce and radish) and observed significant differences, possibly due to the bacterial attachment differences across the plant varieties during production.

Riggio et al. [42] reviewed the various factors which impact the internalization of pathogens during the lab-scale cultivation of leafy vegetables, including microgreens, and noted several studies which identified plant age-specific, plant variety-specific, and pathogen-specific factors that drive endophytic colonization [72,74]. In the present study, minimal differences were observed between the chemical compositions of the sunflower and pea shoot microgreens, with some variation based on the SFCM type used during production. However, with the numerous co-factors, i.e., pathogen type and SFCM type, it would be difficult to pinpoint the impact of a single analyte on pathogen persistence and transfer to the edible portion of the microgreen. Future research should focus on determining how the properties of microgreen varieties influence pathogen survival, as well as potential intervention strategies that do not impact microgreen yield and nutritional value. Likewise, many aspects of microgreen production should be further evaluated in order to characterize food safety risks. The mechanisms of root and plant colonization, as well as the adaptive responses of foodborne pathogens to the plant environment have been evaluated for other plant types [63,75], and characterization of the molecular response of pathogens during microgreen colonization is warranted. In addition, the rhizosphere plant exudates among microgreen varieties may impact pathogen colonization as well [68].

As with any study on the interactions of pathogens and plants, there were a few limitations in this present study. Firstly, maintaining the conditions of the growing environment were challenging, which was primarily due to the low relative humidity during the winter months. These issues resulted in several unsuccessful attempts to grow the microgreens indoors. Future work should limit fluctuations in the growing environment between trials and strive to maintain the environmental parameters implemented in the CEA industry. As previously mentioned, a higher variability of log CFU/g recovery was observed for the pathogen concentrations obtained from the peat samples, which is generally problematic when recovering bacteria during soil sampling [76] and is of little concern for growing mats. This inconsistency could also be related to variable resuscitation of injured bacterial cells; however, no specific methods, such as a pre-enrichment step [77] or a thin agar layer method [78] for the recovery of injured cells, were applied in the present study. Future stud-

ies should characterize the impact of the microgreen cultivation environment on bacterial cell metabolic state.

In addition, the microgreens were watered throughout the cultivation period to maintain an optimal production environment. Previous research has indicated the impact of soil saturation levels on pathogen internalization via the roots [27], stating that the maximum internalization of pathogens such as *E. coli* O157:H7 occurs when the soil is saturated with water in comparison to it being just "moist". It is possible that the daily watering of SFCM facilitated pathogen transfer to the microgreens in the present study, though the effect of the plant variety on pathogen transfer to the edible portion of the microgreen has still been observed regardless of the SFCM saturation. Lastly, the pathogens were inoculated at 5 to 6 log CFU/g of SFCM in the present study; however, pathogens are unlikely to be present in the natural environment at these concentrations. While some studies have reported differences in pathogen uptake by leafy greens under hydroponic growth conditions based on the inoculum level [79], there is a paucity of data related to the impact of the inoculum level on pathogen transfer to microgreens grown in a soil-free cultivation matrix. Thus, this is a future area of research that should be pursued.

## 5. Conclusions

In conclusion, this research further confirms that the plant variety used can impact foodborne pathogen persistence on produce. This work provides valuable data on the fate of these pathogens in commonly grown microgreen varieties cultivated on biostrate and peat SFCM. As more data is generated on the impact of the microgreen variety and SFCM type on pathogen persistence, control measures can be optimized and implemented to reduce risks. Overall, this research has implications for the CEA industry and will help stakeholders characterize the food safety risks in microgreen growing operations that are specific to microgreen variety.

**Supplementary Materials:** The following are available online at https://www.mdpi.com/article/10.3390/horticulturae7110446/s1, Table S1: Chemical composition analysis of SFCM in the presence or absence of sunflower or pea shoot microgreens on day 10 (Trial 2 only).

**Author Contributions:** Conceptualization, G.M.M. and K.E.G.; methodology, W.D., G.M.M., C.A.B., and K.E.G.; validation, W.D. and C.A.B.; formal analysis, W.D. and C.A.B.; investigation, W.D. and C.A.B.; resources, K.E.G.; data curation, W.D. and C.A.B.; writing—original draft preparation, W.D., G.M.M., and C.A.B.; writing—review and editing, W.D., G.M.M., C.A.B., and K.E.G.; visualization, W.D. and C.A.B.; supervision, C.A.B. and K.E.G.; project administration, K.E.G.; funding acquisition, K.E.G. All authors have read and agreed to the published version of the manuscript.

**Funding:** This research received no external funding. This work was supported in part by the National Institute of Food and Agriculture (NIFA), U.S. Department of Agriculture (USDA), Hatch Act.

**Institutional Review Board Statement:** Not applicable.

**Informed Consent Statement:** Not applicable.

**Data Availability Statement:** The data presented in this study are available on request from the corresponding author.

**Acknowledgments:** We thank Jung Ae Lee for guidance on statistical analyses.

**Conflicts of Interest:** The authors declare no conflict of interest.

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
