# Peer review of "Persistence and Transfer of Foodborne Pathogens to Sunflower and Pea Shoot Microgreens during Production in Soil-Free Cultivation Matrix"

_horticulturae, doi:10.3390/horticulturae7110446_

Round 1
Reviewer 1 Report
I do find this work interesting and valuable. The manuscript is written very well. Some suggestions to improve are below.
Line 14 – L. monocytogenes – mentioned for the first time, so it should be in full latin name
Line 163 – which two trial analyses do you mean? Was the experiment repeated two times? I did not find such information in material and methods.
Latin names are in some places in the manuscript written not in italic – f.ex. line 260. Please check whole manuscript and correct.
Author Response
Responses to each reviewer have been included in the attached file.

Reviewer 2 Report
This article aimed to determine bacterial pathogen persistence during microgreen cultivation 12 and transfer from soil-free cultivation matrix (SFCM) to mature microgreens. The authors concluded that plant variety could impact foodborne pathogen persistence on produce, and pathogen transfer to microgreens during production is influenced by SFCM and microgreen variety.
The article is straightforward, and it contains original information.
This article would be improved if the authors clarify or revise the following:
Lines 76-78. Why did the authors use BHI and TSB for inoculation of Listeria and Salmonella, respectively?
Lines 82-91. Have the authors observed any interaction and/or competitive inhibition between Listeria and Salmonella during the incubation for the cultures prepared at Day 0 and following 10 days of microgreen cultivation?
Lines 102 and 180. Clarify either 2.5 cm2 or 6.25 cm2.
Line 104. Clarify bacteria recovery method from samples. Would the vortexing alone without any physical force such as stomaching be enough to detach all bacteria from the samples in to PBS? How about injured cells? Would it not be better to use XLD- and MOX-TSA overlay methods? In addition, enrichment in a non-selective broth medium may recover injured cells better.
Lines 107-108. Would not 80°C for 24h affect the survival level of bacteria?
Lines 116-117. Revise to “sunflower (Helianthus annuus) seeds (Tiensvold Farms, Rushville, NE) ….”
Line 118. Remove “seeds.”
Lines 215- 228, 260, 263. Italicize bacterial species names.
Line 345. Revise to “leafy greens suggests ….”
Author Response

(The authors gave the same response as above.)

Reviewer 3 Report
Very interesting work described clearly and in detail and does not leave much room for comments. What I'm interested in is that you determined the number of bacteria on the surface and even though you weighed 5 g before the determination, you further confirmed the weight of the samples by being dried in an oven at 80ºC for
24 h to determine the sample dry weight and log CFU / g. I don't understand why? Have you considered incorporating sonication to separate the adherent bacteria or bacteria in the biofilm from your samples?
Author Response

(The authors gave the same response as above.)

Reviewer 4 Report
Section Introduction Please provide more information about S. Javiana selection (make more clearly about the necessity to monitor its presence). Section, Materials and Methods Please consider the use of a flow chart (in terms of a figure or a table, similar to Figure 1) with all sampling information and parameter analyzed in each step. This will provide a rapid understanding of the work that was performed.Author Response
Responses to each reviewer have been included in the attached file.
